# *Saxifraga spinulosa*-Derived Components Rapidly Inactivate Multiple Viruses Including SARS-CoV-2

**DOI:** 10.3390/v12070699

**Published:** 2020-06-28

**Authors:** Yohei Takeda, Toshihiro Murata, Dulamjav Jamsransuren, Keisuke Suganuma, Yuta Kazami, Javzan Batkhuu, Duger Badral, Haruko Ogawa

**Affiliations:** 1Research Center for Global Agromedicine, Obihiro University of Agriculture and Veterinary Medicine, 2-11 Inada, Obihiro, Hokkaido 080-8555, Japan; ytakeda@obihiro.ac.jp (Y.T.); k.suganuma@obihiro.ac.jp (K.S.); 2Department of Pharmacognosy, Tohoku Medical and Pharmaceutical University, 4-1-1 Komatsushima, Aoba-ku, Sendai 981-8558, Japan; murata-t@tohoku-mpu.ac.jp; 3Department of Veterinary Medicine, Obihiro University of Agriculture and Veterinary Medicine, 2-11 Inada, Obihiro, Hokkaido 080-8555, Japan; duuya.dj@gmail.com (D.J.); sketch1111kazami@gmail.com (Y.K.); 4National Research Center for Protozoan Diseases, Obihiro University of Agriculture and Veterinary Medicine, Inada, Obihiro, Hokkaido 080-8555, Japan; 5School of Engineering and Applied Sciences, National University of Mongolia, P.O.B-617/46A, Ulaanbaatar 14201, Mongolia; jbatkhuu@hotmail.com; 6Mongolian University of Pharmaceutical Sciences, Ulaanbaatar 18130, Mongolia; badral_duger@yahoo.com

**Keywords:** medicinal herb, pandemic, pyrogallol-enriched fraction, SARS-CoV-2, *Saxifraga spinulosa*, virus disinfectant

## Abstract

Novel severe acute respiratory syndrome coronavirus 2 (SARS-CoV-2), influenza A virus (IAV), and norovirus (NV) are highly contagious pathogens that threaten human health. Here we focused on the antiviral potential of the medicinal herb, *Saxifraga*
*spinulosa* (*SS*). Water-soluble extracts of *SS* were prepared, and their virus-inactivating activity was evaluated against the human virus pathogens SARS-CoV-2 and IAV; we also examined virucidal activity against feline calicivirus and murine norovirus, which are surrogates for human NV. Among our findings, we found that *SS*-derived gallocatechin gallate compounds were capable of inactivating all viruses tested. Interestingly, a pyrogallol-enriched fraction (Fr 1C) inactivated all viruses more rapidly and effectively than did any of the component compounds used alone. We found that 25 µg/mL of Fr 1C inactivated >99.6% of SARS-CoV-2 within 10 s (reduction of ≥2.33 log_10_ TCID_50_/mL). Fr 1C resulted in the disruption of viral genomes and proteins as determined by gel electrophoresis, electron microscopy, and reverse transcription–PCR. Taken together, our results reveal the potential of Fr 1C for development as a novel antiviral disinfectant.

## 1. Introduction

Severe acute respiratory syndrome coronavirus 2 (SARS-CoV-2), influenza A virus (IAV), and norovirus (NV) are highly contagious pathogens that threaten the health of the worldwide community. SARS-CoV-2 is an enveloped RNA virus (genus *Betacoronavirus*, subgenus *Sarbecovirus*) that was first identified in December 2019 in patients with infectious pneumonia in Wuhan, China [1,2]. SARS-CoV-2 has since spread throughout the world; the World Health Organization (WHO) declared this outbreak to be a pandemic on March 11, 2020. The worldwide incidence of SARS-CoV-2 infection has continued to rise; as of June 19, 2020, more than 8.3 million infections and ~450,000 deaths have been reported [3]. IAV is also an enveloped RNA virus (Family Orthomyxoviridae) and is the agent responsible for seasonal infections of varying severity. Four recent pandemics have resulted from the introduction of novel strains of IAV; within the past 100 years, ~500,000 to 50 million deaths have been attributed to these pandemics [4]. NV is non-enveloped RNA virus (Family Calciviridae) that is the etiologic agent of acute gastroenteritis; more than 120 million individuals were infected with this virus in 2010 [5]. Young children and the elderly are at particularly high risk for severe disease and death in response to NV [6]; many NV outbreaks have been identified in long-term care facilities for elderly adults [7].

Effective prophylaxis and antiviral therapies are both important factors that are critical for managing highly contagious viruses. Several drugs, including the nucleoside analog, remdesivir, the antimalarial drug, chloroquine, and the serine protease inhibitor, camostat mesylate, showed activity against SARS-CoV-2 in experiments performed in vitro [8,9]. Although a compassionate-use study with remdesivir suggested some initial clinical improvement [10], further strictly-controlled trials have not yet been completed. There are currently no drugs available to treat NV, and while several anti-IAV drugs are currently available, the problem of emerging drug-resistant viruses has not been solved [11,12].

Given this situation, one of the most practical and effective protective measures could include the implementation of antiviral disinfectants. The WHO recently documented the effectiveness of alcohol-based solutions as disinfectants for the prevention of transmission of SARS-CoV-2 [13]; these solutions are also effective against IAV. However, the current shortage of existing disinfectants, most notably in medical institutions, has become a serious problem given the ongoing SARS-CoV-2 pandemic. The current shortage of existing disinfectants may result in increased transmission of not only SARS-CoV-2 but also of other contagious viruses including IAV and NV. Alcohol-based agents are unsuitable for use in infants, individuals with aldehyde dehydrogenase-deficiency, and children with skin damage. Long-term, continuous use of alcohol solutions also can lead to unacceptable skin irritation [14]. Of note, NV is alcohol-resistant; sodium hypochlorite (chlorine bleach) is recommended for antiviral disinfection; this agent is not suitable for topical use and can be corrosive to the environment. Hence, there is an ongoing and increasing demand for safe and effective antiviral disinfectants which can be used continuously with no adverse effects.

*Saxifraga spinulosa* (*SS*) Adams (family, Saxifragaceae) is a perennial herbaceous plant that is distributed widely in eastern continental Asia. Flavonoids, phenolic acids, terpenoids, and phytosterols have all been reported as chemical constituents of *Saxifraga* plants; compounds that feature a pyrogallol B-ring as well as galloyl moieties have been identified as constituents of *SS*-derived extracts [15]. Furthermore, many of the compounds isolated from *SS* have DPPH (2,2-diphenyl-1-picrylhydrazyl) radical-scavenging activities and/or can inhibit *Babesia* and *Theileria* parasites [15]. In Asian countries, *Saxifraga* species are used as medicinal herbs for treatment of various types of disorders. With respect to antiviral activities, Zuo et al. [16] reported that *Saxifraga melanocentra* inhibited the activity of hepatitis C virus serine protease. Here, we assessed the virucidal activities of *SS*-derived components against SARS-CoV-2 and IAV, as well as feline calicivirus (FCV) and murine NV (MNV), which are surrogates for human NV.

## 2. Materials and Methods

### 2.1. Viruses and Cells

Human IAV (H1N1; A/Puerto Rico/8/1934 strain: ATCC^®^ Catalog No. VR-95^TM^) was purchased from ATCC (Manassas, VA, USA) and propagated in allantoic fluid of 10-day-old embryonated chicken eggs. For the preparation of purified IAV, sucrose gradient ultracentrifugation was performed as previously described [17]. The protein concentration of purified IAV was measured using the Micro BCA protein assay kit (Thermo Fisher Scientific Inc., Waltham, MA, USA). Madin–Darby canine kidney (MDCK) cells were kindly provided by H. Nagano (Hokkaido Institute of Public Health, Sapporo, Japan). The composition of MDCK cell growth medium and IAV growth medium were as previously described [18]. FCV (F9 strain) and Crandell-Rees feline kidney (CRFK) cells were kindly provided by Dr. K. Maeda (Yamaguchi University, Yamaguchi, Japan). CRFK cells were grown and passaged in the medium used for culture and growth of the MDCK cells. After inoculation with FCV, CRFK cells were cultured in viral growth medium: Dulbecco’s Modified Eagle’s medium (DMEM; Nissui Pharmaceutical Co., Ltd., Tokyo, Japan) supplemented with 1% fetal bovine serum, 2 mM l-glutamine (Wako Pure Chemical Industries, Ltd., Osaka, Japan), 0.15% NaHCO_3_ (Wako Pure Chemical Industries, Ltd.), 2 µg/mL amphotericin B (Bristol-Myers Squibb Co., New York, NY, USA), and 100 µg/mL kanamycin (Meiji Seika Pharma Co., Ltd., Tokyo, Japan). MNV (S7 strain) was kindly provided by Dr. Y. Tohya (Nihon University, Tokyo, Japan). The murine leukemia macrophage cell line RAW 264.7 was provided by the RIKEN BRC (Tsukuba, Japan) through the National Bio-Resource Project of the Ministry of Education, Culture, Sports, Science and Technology (MEXT), Japan. RAW 264.7 cells were cultured and passaged in the medium used for the MDCK cells. After MNV inoculation, RAW 264.7 cells were cultured viral growth medium as described above for the CRFK cells. For the preparation of purified FCV or MNV, cell supernatant containing propagated FCV or MNV was subjected to ultracentrifugation (10,000× *g*, 3 h), through 30% sucrose (Nacalai Tesque Inc., Kyoto, Japan) in ultracentrifuge tubes (Hitachi Koki Co., Ltd., Tokyo, Japan). The virus pellet was resuspended in phosphate-buffered saline (PBS). IAV, FCV, and MNV were handled in the biosafety level 2 facility. SARS-CoV-2 (JPN/TY/WK-521 strain) and transmembrane protease, serine 2 (TMPRSS2)-expressing VeroE6 (VeroE6/TMPRSS2) cells [19] were kindly provided by National Institute of Infectious Diseases (Tokyo, Japan). For passaging, VeroE6/TMPRSS2 cells were cultured in the growth medium described for the MDCK cells with the addition of G418 disulfate aqueous solution (Nacalai Tesque Inc.) at 500 µg/mL. After inoculation with SARS-CoV-2, VeroE6/TMPRSS2 cells were cultured in viral growth medium composed of DMEM supplemented with 1% fetal bovine serum, 2 mM l-glutamine, 0.15% NaHCO_3_, and 100 µg/mL kanamycin. SARS-CoV-2 was handled in the biosafety level 3 facility.

### 2.2. SS-Derived Samples

Extraction and isolation of plant materials as well as structural determination procedures were as described in a previous report [15]. The compounds isolated from each fraction and their respective yields are included in Table 1. The aqueous extract (96.1 g) was subjected to chromatography on a Diaion HP-20 column (70 × 470 mm), with elution performed using water (Fr 1A, 45.4 g), methanol –water (1:4, *v*/*v*) (Fr 1B, 8.3 g), methanol–water (2:3, v/v) (Fr 1C, 19.4 g), methanol–water (3:2, *v*/*v*) (Fr 1D, 17.2 g), methanol–water (9:1, *v*/*v*) (Fr 1E, 3.4 g), methanol (Fr 1F, 219 mg; Appendix A). Compounds **1**–**8** were obtained from 5.0 g of Fr 1C; compounds **9**–**17** were obtained from 5.0 g of Fr 1D; compounds **18**–**29** were obtained from 3.3 g of Fr 1E (Table 1). Each extract, fraction, and compound was dissolved in CultureSure dimethyl sulfoxide (DMSO) (Wako Pure Chemical Industries, Ltd.) (1 mg/100 μL), and diluted to PBS to use for the assay.

### 2.3. Screening for Virucidal Activity

The *SS*-derived samples were added to solutions containing IAV, FCV, MNV, or SARS-CoV-2. The concentrations of *SS*-derived samples were 25, 25, 100, and 25 µg/mL in the mixtures containing IAV, FCV, MNV, and SARS-CoV-2, respectively. DMSO was used as the diluent control. The viral titers of the control mixtures (mixtures containing DMSO) were 10^3.25-6.75^ 50% tissue culture infective dose (TCID_50_)/mL. Solutions with virus and *SS*-derived components were incubated at 25 °C for varying periods of time, from 10 s to 48 h. At the end of the incubation period, 10-fold serial dilutions were prepared for inoculation into respective cell line hosts. Three days after IAV, FCV, and MNV inoculation or 2 days after SARS-CoV-2 inoculation, viral titers were evaluated by observing the cytopathic effects on the respective cell lines. The log_10_ TCID_50_/mL was calculated using the Behrens–Kärber method [20]. The differences in titer in virus solutions treated with each *SS*-derived sample vs. DMSO control were evaluated.

### 2.4. Sodium Dodecyl Sulfate–Polyacrylamide Gel Electrophoresis (SDS–PAGE)

The *SS*-derived samples were added to solutions of purified IAV, and FCV, and MNV. The concentrations of *SS*-derived samples were 250, 250, and 1000 µg/mL in the mixture containing purified IAV (8.25 log_10_ TCID_50_/mL), FCV (7.25 log_10_ TCID_50_/mL), and MNV (7.25 log_10_ TCID_50_/mL), respectively. Fr 1C was incubated with cell supernatant containing SARS-CoV-2. The concentration of Fr 1C was 250 µg/mL in the mixture containing SARS-CoV-2 (4.75 log_10_ TCID_50_/mL). DMSO was utilized as a diluent control. *SS*-derived samples and viruses were placed at 25 °C for 48 h (IAV), 18 h (FCV), 48 h (MNV), and 48 h (SARS-CoV-2), and then subjected to SDS–PAGE. After the reaction time, these mixtures were mixed with the equal volume of 2× SDS sample buffer with and/or without 2-mercaptoethanol (2-Me; Wako Pure Chemical Industries, Ltd.). The SDS–PAGE samples were heated to 100 °C for 2 min and then loaded onto a 12% polyacrylamide gel for electrophoresis. Precision Plus Protein™ All Blue Prestained Protein Standards or Unstained Protein Standards (Bio-Rad Laboratories Inc., Hercules, CA, USA) were included as molecular weight markers. The gel was stained with Coomassie brilliant blue (CBB) Stain One (Nacalai Tesque Inc.) and images were taken using the LAS-3000 Imaging System (Fujifilm Co., Tokyo, Japan) with white light transillumination. For western blotting (WB) to detect IAV HA protein, proteins were transferred from the acrylamide gel to a polyvinylidene difluoride membrane (Bio-Rad Laboratories Inc.) that was probed with rabbit anti-H1N1 (A/Puerto Rico/8/34) HA polyclonal antibody (Catalog No. 11684-T62, Sino Biological Inc., Beijing, China). The membrane was then probed with mouse anti-rabbit IgG peroxidase conjugate (Catalog No. A1949, Clone: RG-96, Sigma-Aldrich, Inc., Saint Louis, MO, USA) and developed with ECL Prime Western Blotting Detection Reagent (GE Healthcare Ltd., Chicago, IL, USA) and chemiluminescence detection using the LAS-3000 Imaging System. For WB to detect IAV NA protein, membranes were probed sequentially with rabbit anti-H1N1 NA (Catalog No. GTX125974, Genetex Inc., Irvine, CA, USA) and mouse anti-rabbit IgG peroxidase conjugate, followed by development as described. MNV VP1 protein was detected using mouse anti-MNV-1 monoclonal antibody (Catalog No. MABF2097, clone: 5C4.10, Merck & Co., Inc., Kenilworth, NJ, USA) and horseradish peroxidase-conjugated goat anti-mouse IgG2b (Catalog No. M32407, Thermo Fisher Scientific Inc.). SARS-CoV-2 S2 subunit protein was detected using rabbit anti-spike/S2 polyclonal antibody (Catalog No. 40590-T62, Sino Biological Inc.) and mouse anti-rabbit IgG peroxidase conjugate. SARS-CoV-2 nucleoprotein (NP) was detected using rabbit anti-NP monoclonal antibody (Catalog No. 40143-R019, Sino Biological Inc.) and mouse anti-rabbit IgG peroxidase conjugate.

### 2.5. Hemagglutination (HA) and Neuraminidase (NA) Assays

The *SS*-derived samples or DMSO diluent controls were added to a PBS containing purified IAV. The concentration of *SS*-derived samples was 250 µg/mL in the mixture containing purified IAV (8.25 log_10_ TCID_50_/mL). The mixture was incubated at 25 °C for 48 h. HA assay and NA assay were conducted according to the WHO manual on influenza diagnosis and surveillance [21].

### 2.6. RT–PCR

The *SS*-derived samples were added to solutions of purified IAV, FCV, and MNV. The concentration of *SS*-derived samples was 250 µg/mL in the mixture containing purified IAV (8.25 log_10_ TCID_50_/mL), FCV (7.25 log_10_ TCID_50_/mL), and MNV (7.25 log_10_ TCID_50_/mL). Fr 1C was incubated with cell supernatant containing SARS-CoV-2. The concentration of Fr 1C was 250 µg/mL in the mixture containing SARS-CoV-2 (4.75 log_10_ TCID_50_/mL). DMSO was utilized as a diluent control. The mixture was incubated at 25 °C for 48 h, 12 h, 48 h respectively for IAV, FCV and MNV, and 48 h for SARS-CoV-2. After the reaction time, virion RNA was isolated using ISOGEN-LS (Nippon Gene, Tokyo, Japan); the concentration of RNAs isolated from all samples was adjusted to 100 µg/mL. RNA (500 ng) was reverse transcribed using FastGene cDNA Synthesis 5× ReadyMix OdT (NIPPON Genetics Co, Ltd., Tokyo, Japan) and reverse transcription–polymerase chain reaction (RT–PCR) was performed using GoTaq^®^ Green Master Mix (Promega Co., Madison, WI, USA). The sequence of primers used in this study and each PCR condition are presented in the Appendix A. The direct effect of Fr 1C on isolated IAV RNA was also evaluated. Fr 1C (250 µg/mL) was added to a solution of RNA (250 µg/mL) from purified IAV, with double distilled water (DDW) and DMSO included as controls. The reactions were incubated at 25 °C for 18 or 48 h. The reactions were then diluted 20 times with PBS and 20 µg/mL of RNA isolated from purified MNV was added to serve as an intact control RNA to assess the impact of DMSO or Fr 1C on enzymatic reactions associated with RT–PCR. To minimize the impact of DMSO or Fr 1C on the intact control RNA, the mixtures were reverse transcribed immediately after the addition of the intact control RNA; RT–PCR to detect both IAV RNA and intact control RNA were performed.

### 2.7. Transmission Electron Microscopy

Fr 1C (250 µg/mL) or DMSO diluent was added to a PBS containing purified IAV (200 µg/mL, 5.75 log_10_ TCID_50_/mL). The reactions were incubated at 25 °C for 48 h. Samples were prepared for electron microscopy as previously described [17]. The samples were evaluated using a transmission electron microscope (HT7700; Hitachi High-Tech Co., Tokyo, Japan). To determine the percentage of intact virus in each sample, 0.52 µm^2^ fields in which more than 3 viral particles were present were selected and the ratio of the intact virus to the total number of virus was calculated. The virus keeping its particle structure and having clear spike proteins was judged as the intact virus. Thirty fields were observed in each DMSO- and Fr 1C-treated sample and the percentages between the two sample groups were compared.

### 2.8. Statistical Analysis

Student’s *t* test was performed to determine statistically significant differences between DMSO control groups and each *SS*-derived sample group for the initial antiviral screening test. Student’s *t* test was also performed between DMSO control group and each *SS*-derived sample group for the viral and HA titers used to evaluate purified IAV. Student’s *t* tests were also performed to identify statistically significant differences between the DMSO control group and the Fr 1C-treated sample group evaluated by electron microscopy. *p* values < 0.05 were used to determine statistical significance.

## 3. Results

### 3.1. Identification of SS-Derived Components with Virucidal Activity against IAV, FCV, and MNV

We performed screening to identify *SS*-derived virucidal components. The first evaluation focused on the virucidal activities of the aqueous extract and Fr 1A–F (Appendix A) [15] against IAV, FCV, and MNV. Fr 1A–F were fractionated from aqueous extract (Appendix A). Virus preparations that were not gradient-purified were used in this analysis. All fractions (25 µg/mL) except Fr 1A promoted significant IAV-inactivating activity after 6 h; the aqueous extract and Fr 1C–E fractions were capable of inactivating virus within 1 min. The virucidal activity of Fr 1C was the most potent of this set; incubation with Fr 1C resulted in >99.98% inactivation of IAV (i.e., a reduction ≥3.83 log_10_ TCID_50_/mL; Table 2A). All fractions at 25 µg/mL showed significant activity against FCV after 6 h of incubation. All fractions except Fr 1A inactivated FCV within 1 min; >99.79% of FCV was inactivated within this time frame (reduction ≥2.67 log_10_ TCID_50_/mL; Table 2B). None of the fractions had virucidal activity against MNV when evaluated at 25 µg/mL (Appendix A). However, at 100 µg/mL, the aqueous extract and Fr 1B–E were capable of inactivating MNV after 6 h of incubation; Fr 1C and 1D were capable of inactivating MNV within 1 min. Overall, MNV inactivation of >98.52% was observed in response to these fractions (reduction ≥1.83 log_10_ TCID_50_/mL; Table 2C).

We then evaluated the virus-inactivating activity of the *SS*-derived compound Nos. **1**–**29** (Table 1, Appendix A [15]). Compound Nos. **2**–**4** (25 µg/mL) had virucidal activity against both of IAV and FCV at 1 min, with >93.24% inactivation of IAV and FCV (reduction ≥1.17 log_10_ TCID_50_/mL; Table 3A,B). Compound No. **2** (gallocatechin-3-*O*-gallate) and No. **4** (epigallocatechin-3-*O*-gallate), each at 100 µg/mL, had significant MNV-inactivating activity at 6 h, although neither were effective at 10 min (Table 3C). Overall, the Fr 1C- and Fr 1D-derived compounds had notable virucidal activities; the virucidal impact of the Fr 1C compounds were overall stronger than those derived from of Fr 1D. These results indicate that Fr 1C and Fr 1D contain multiple virucidal compounds that promote potent and rapid virus inactivation more effectively together than as single compounds.

### 3.2. Impact of SS-Derived Fractions on IAV Proteins and Genome

We first analyzed the antiviral activity of *SS*-derived fractions against high-titer preparations of purified IAV. The viral titers detected after treatment with the aqueous extract or Fr 1B–D decreased below the detection limit after 48 h (Figure 1A). We then analyzed the impact of *SS*-derived fractions on virus proteins under these conditions; the IAV virions treated with each fraction were evaluated by SDS–PAGE. CBB staining revealed a decreased intensity of some virus protein bands in response to treatment with several of the fractions. Of particular note, Fr 1C-treatment resulted in the strongest reduction in band intensity, which we tentatively identified as the viral HA multimer, HA0, HA1 and HA2 proteins based on their apparent molecular weights. Interestingly, intensities of the bands tentatively identified as NP and M1 proteins were not affected in response to treatment with *SS*-derived fractions (Figure 1B). To assess the impact of the *SS*-derived fractions specifically on the viral HA proteins, WB was performed using a first antibody that detects IAV HA protein. Consistent with the results of CBB staining, treatment with Fr 1C resulted in the strongest reduction in band intensity of the HA proteins (Figure 1C). A HA assay was performed to evaluate protein function. As anticipated, treatment with Fr 1C resulted in a substantial decrease in HA activity (Figure 1D). We then performed WB to determine the impact of *SS*-derived fractions on the viral NA protein. Treatment with the *SS*-derived aqueous extract or Fr 1B–1E resulted in the full disappearance of NA-specific band on WB (Figure 1E); IAVs treated with all fractions except for Fr 1A lost all NA activity (Figure 1F).

We next assessed the impact of *SS*-derived fractions on IAV genome by RT–PCR; primers were designed to amplify three different regions of the IAV M gene and to generate amplicons of distinct lengths. The cDNA derived from IAV treated with each fraction was PCR-amplified. As anticipated, no PCR products were detected from treated IAVs save for those treated with Fr 1A (Figure 2A). To evaluate direct impact of Fr 1C on the IAV genome, RT–PCRs were performed with primers that amplify partial sequence of the IAV M gene after 18 and 48 h incubation with Fr 1C; DDW was used as negative control and DMSO as a solvent control (Figure 2B-IAV RNA). The amount of IAV PCR product detected on the gel decreased in response to Fr 1C-treatment in a time dependent manner (Figure 2C). By contrast intact control RNA that was added in the IAV RNA containing solution immediately prior to the RT–PCR reaction was fully amplified in all three samples (Figure 2B-Intact control RNA and 2C). These results indicated that DMSO and Fr 1C have no inherent capacity to interfere with enzyme reactions related to RT–PCR; these results confirm that Fr 1C acts directly to destroy the IAV genome template.

We assessed morphological changes of IAV particles treated with Fr 1C by electron microscopy. The virion number was substantially decreased by treatment with Fr 1C (Appendix A). DMSO-treated IAVs maintained normal virion structure; however, Fr 1C-treated IAVs included numerous abnormal structures including disrupted particles (Figure 3A: black arrowhead) and rough spike proteins (Figure 3A: white arrowheads). The fraction of intact viruses underwent a significant decreased in response to Fr 1C-treatment (Figure 3B). These results indicate that Fr 1C inactivates IAV by disrupting HA and NA proteins as well as the virus genome.

### 3.3. Impact of SS-Derived Fractions on FCV and MNV Proteins and Genomes

To analyze the impact of *SS*-derived fractions on proteins and the genome of purified FCV, SDS–PAGE and RT–PCR were performed after 18 h and 12 h, respectively. The viral titer of purified FCV treated with each fraction was also evaluated (Appendix A). CBB staining of the polyacrylamide gel revealed a band of 63 kDa, which was tentatively identified as virus VP1 protein; the intensity of this band decreased in response to treatment with several of the *SS*-derived fractions, most notably in response to Fr 1C (Figure 4A). The cDNA derived from the FCV virions treated with each fraction was amplified with PCR; primers were designed to amplify 264 bp region on VP1 gene. PCR products detected on the gel had decreased or were not detected in response to all *SS*-derived fractions except Fr 1A (Figure 4B). The impact of *SS*-derived fractions on proteins and the genome of purified MNV was also analyzed after 48 h, and the viral titers of purified MNV treated with each fraction were measured (Appendix A, D). WBs performed to detect MNV VP1 protein revealed that the specific band decreased in response to treatment with Fr 1C and 1D (Figure 4C). Then cDNA derived from the MNV treated with each fraction was amplified with PCR; primers were designed to amplify 721 bp region on the 3′ terminus of ORF1 and the 5′ terminus of ORF2 [22]. The PCR product was not detected in response to treatment with the aqueous extract or to Fr 1C–E (Figure 4D).

### 3.4. Identification of SS-Derived Fractions that Promote Rapid Inactivation of SARS-CoV-2

We found that the aqueous extract and Fr 1B–E (all at 25 µg/mL) had significant virucidal activity against SARS-CoV-2 within 1 min of exposure; >99% of the virus was inactivated by components within these fractions (i.e., reduction of ≥2 log_10_ TCID_50_/mL). We found that >99.53% of the virus was inactivated by Fr 1C within 10 s of exposure (reduction of ≥2.33 log_10_ TCID_50_/mL; Table 4A). We also evaluated the *SS*-derived single compounds for virucidal activity against SARS-CoV-2. Fr 1C compounds No. **2** (gallocatechin-3-*O*-gallate) and No. **4** (epigallocatechin-3-*O*-gallate) each at 25 µg/mL were capable of inactivating SARS-CoV-2 within 1 min of exposure; interestingly, these activities were not as robust as those of the aqueous extract and Fr 1B–E fractions (Table 4B). These results characterize the virtually instantaneous and potent virucidal activity of Fr 1C against SARS-CoV-2.

### 3.5. Impact of SS-Derived Fractions on SARS-CoV-2 Proteins and Genome

To analyze the impact of Fr 1C on SARS-CoV-2 proteins and genome, WB and RT–PCR were performed after 48 h. At the same time, the viral titer of SARS-CoV-2 treated with Fr 1C was analyzed (Appendix A). WB was performed to detect S2 subunit, a component of the spike protein of SARS-Cov-2, and NP revealed that the specific band decreased in response to treatment with Fr 1C (Figure 5A,B). The cDNA derived from SARS-CoV-2 treated with Fr 1C was amplified with PCR; primers were designed to amplify 158 bp region on N gene. No PCR products were detected in response to treatment with Fr 1C (Figure 5C).

## 4. Discussion

In this study, we identified *SS*-derived compounds which inactivate multiple viruses that are of critical public health concern, i.e., IAV, NV, and SARS-CoV-2. The fractions from *SS* with prominent virucidal activity were those that included compounds with pyrogallol and or catechol moieties. The gallocatechin gallate compounds (No. **2** and No. **4**) were very effective at inactivating all four virus species evaluated here; as such, the structures of these two compounds may provide critical information toward the design of broadly effective antiviral disinfectants (Table 3 and Table 4B). Other active compounds also have a pyrogallol B-ring within the flavonoid skeleton and/or feature galloyl moieties within their structures. Our studies suggest that the pyrogallol B-ring contributed to the virucidal activity observed here to a greater extent than did the catechol B-ring. For example, compound No. **3** and No. **13** were more effective than compound No. **23** and No. **25** (Table 3). Although we were not able to define specific structure-function relationships based on the findings of these 29 compounds, the results overall point to the contributions of galloyl groups and of the pyrogallol moieties as key virucidal elements. Compounds that feature pyrogallol or catechol moieties have been characterized as antioxidants [23]. Likewise, tannins with multiple hydroxyl groups have astringent action due to interactions of the phenol moieties with proteins [24]. Among the 29 compounds screened here, those with a catechin skeleton is worth noting. Catechin derivatives including epigallocatechin gallate (EGCG, No. **4** in the present study) are among the main constituents of green tea (*Camellia sinensis*); catechins have characterized antiviral activity against a wide spectrum of virus pathogens [25,26], including influenza [27,28]. With respect to SARS-CoV2, our studies contribute important primary knowledge and may provide a larger understanding of potential mechanisms of antiviral efficacy.

Some of the *SS*-derived fractions, most notably the pyrogallol-enriched fraction Fr 1C showed higher virucidal activities than each single compound; e.g., Fr 1C eliminated >99.53% of the SARS-CoV-2 in 10 s of administration (Table 2, Table 3, and Table 4). Among the reasons for this effect, it is possible that there are unknown highly potent virucidal compounds within Fr 1C that are present in low concentration and thus have eluded biochemical characterization. Likewise, the combination of the active compounds results in a synergistic antiviral effect. Nagle et al. [29] discussed this second hypothesis in their review of the biological functions of EGCG; they suggest that synergistic effects may explain some of the discrepancies between antiviral findings identified in vitro and in clinical studies. For example, an in vitro analysis revealed that the active concentration of a single catechin identified in green tea is far above the concentration in vivo after intake of green tea extract, although clinical studies attest to its effectiveness. As such, our studies designed to evaluate the nature of this effect are ongoing.

In our evaluation of IAV inactivation by Fr 1C, the results of SDS–PAGE and the HA and NA tests suggested a mechanism of action associated with disruption of both HA and NA proteins (Figure 1 and Figure 3A: white arrowhead). These results are consistent with those of previous reports that characterized the role of green tea catechins, including EGCG, in promoting HA and NA inhibition [30,31,32,33]. The molecular docking simulations suggested that catechins may bind to the amino acid residues that facilitate HA-host receptor binding and the enzymatic activity of NA [30,33]. The impact of Fr 1C treatment was not limited to the virus HA and NA proteins. Electron microscopy revealed that the number of IAV particle was substantially decreased in response to Fr 1C (Appendix A) and that some virus particles were disrupted (Figure 3A: black arrowhead). Fr 1C treatment also resulted in the disruption of multiple virus proteins and genomes (Figure 2, Figure 4, and Figure 5). The wide spectrum antiviral functions of catechins as already mentioned above may be features underlying the broad virucidal activity of Fr 1C observed in our study.

We also show here that Fr 1C was capable of promoting very rapid virus inactivation. This property would be a critical feature of any novel antiviral disinfectants. Among one of the most interesting of our points, the rapid rate of Fr 1C-mediated virus inactivation can be observed against enveloped viruses including IAV and SARS-CoV-2 and also against non-enveloped viruses including FCV and MNV; this is of particular note, as neither FCV nor MNV can be deactivated by alcohol-based solutions. The pandemic SARS-CoV-2 causes severe symptoms with high mortality in elderly persons; outbreaks in nursing homes and hospitals have been reported. The elderly population is also at high risk for severe infection with IAV and NV. The practical use of an antiviral disinfectant solution developed from Fr 1C for use in nursing homes and hospitals may be effective for the control of multiple contagious viruses. Moreover, the Fr 1C may be useful for not only as a disinfectant but also as applied to other antiviral protective measures. For example, Oxford et al. [34] reported the effective use of a mask containing green tea extract for prevention of IAV infection. In other reports, Ide et al. [27,35] demonstrated the effectiveness of gargling with green tea or its extracts for the prevention of IAV infection; these findings were subjected to a meta-analysis that included five independent clinical research trials. An agent based on the compounds detected in Fr 1C may be useful in these antiviral strategies as well. In addition, although our efforts here focused on Fr 1C as a preventive measure, we are also planning to evaluate is impact on virus replication after infection has been initiated.

## 5. Conclusions

In conclusion, our findings reveal critical virucidal activity of compounds derived from the herbal plant, *S. spinulosa*; *SS*-derived components have virucidal activity against SARS-CoV-2, IAV, as well as surrogate viruses of human NV. Of particular note, Fr 1C, which is enriched with pyrogallol compounds, promoted the profound disruption of virus genomes and proteins. We anticipate that these findings will generate interest into the potent virucidal activities of Fr 1C, which may contribute to the prevention of virus transmission under our current worldwide pandemic crisis.

## Figures and Tables

**Figure 1 viruses-12-00699-f001:**
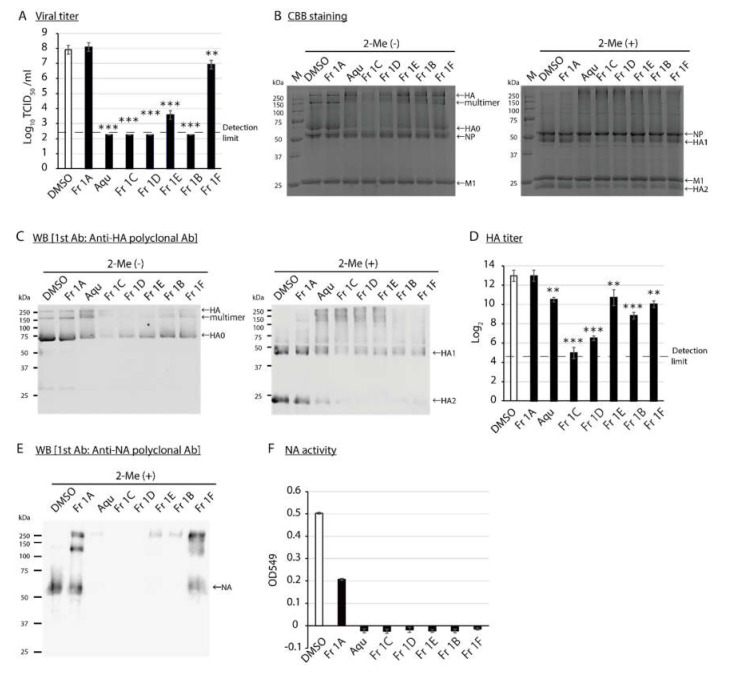
Determination of the impact of *SS*-derived fractions on IAV proteins. (**A****–F**) DMSO control and the *SS*-derived fractions were added to solutions containing purified IAV; reactions proceeded at 25 °C for 48 h. Aqu: Aqueous extract. (**A**) The viral titer of each reaction was evaluated. Error bars indicate mean ± SD; n = 3 per group. (**B**) The images to the left and right are the results of CBB staining of SDS-samples without 2-Me and with 2-Me, respectively. M: Marker. (**C**) The images to the left and right are the results of WB to detect IAV HA proteins. (**D**) The HA titer was evaluated. Error bars indicate mean ± SD; n = 3 per group. (**E**) The image is the result of WB to detect IAV NA protein. (**F**) The NA activity was evaluated. Error bars indicate mean of triplicate measurement of same sample ± SD. (**A,D**) Student’s *t* test was performed to analyze statistical difference; ** *p* < 0.01, *** *p* < 0.001. (**B,C,E,F**) The results are representatives of two individual experiments.

**Figure 2 viruses-12-00699-f002:**
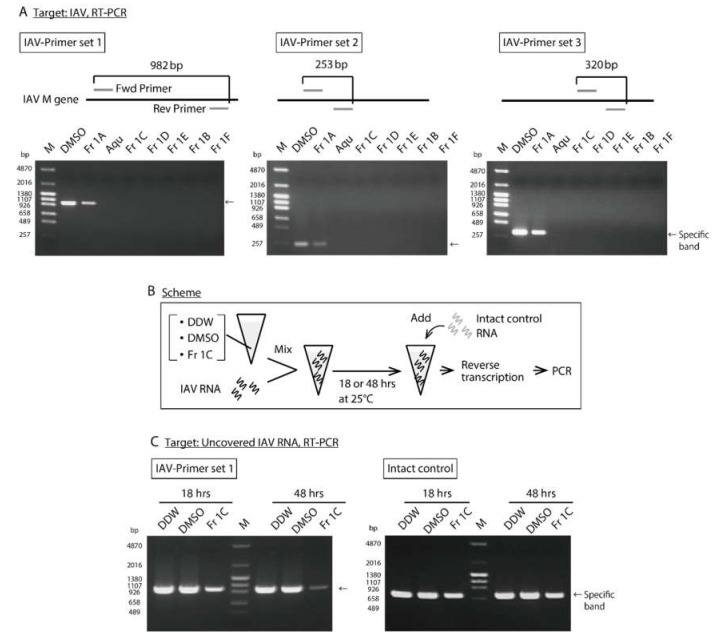
Analysis of the impact of *SS*-derived fractions on IAV genome. (**A**) DMSO and the *SS*-derived fractions were added to solutions containing purified IAV; reactions proceeded for 25 °C for 48 h and evaluated by RT–PCR. The images are the results of RT–PCR using IAV-Primer sets 1, 2, and 3 which amplify 982 bp, 253 bp, and 320 bp regions on IAV M gene, respectively. M: Marker, Aqu: Aqueous extract. (**B,C**) Scheme of the experimental procedure is shown (**B**). DDW, DMSO, and Fr 1C were added to RNA isolated from purified IAV. After the reaction time, the RNA isolated from purified MNV was added which were then subjected immediately to RT–PCR. (**C**) The image on the left is the result of RT–PCR using IAV-Primer set 1. The image on the right depicts the results of RT–PCR using MNV-F1 and -R1 primers which amplify 721 bp region on MNV gene. (**A,C**) The results are representatives of two individual experiments.

**Figure 3 viruses-12-00699-f003:**
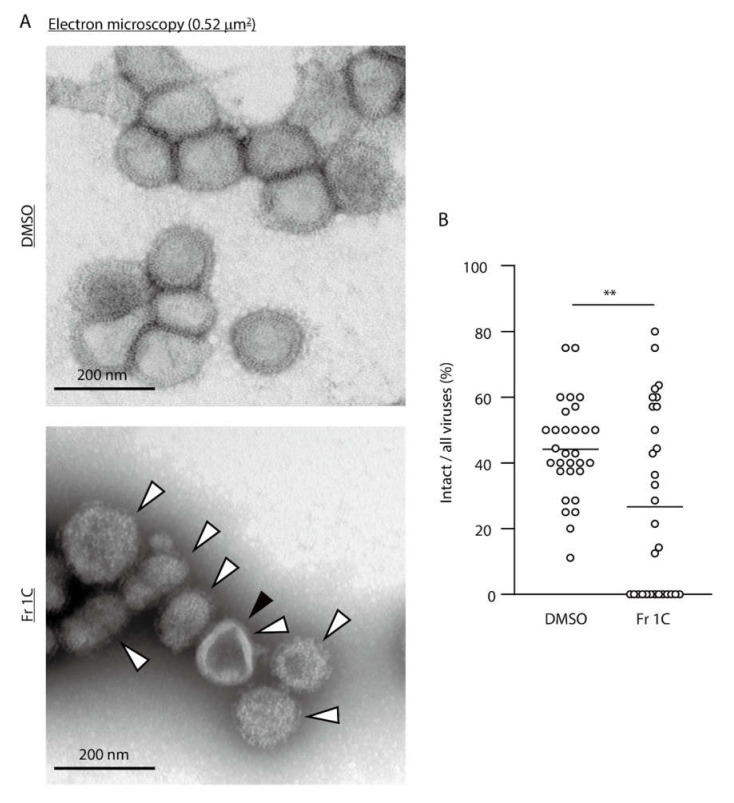
Electron microscopic evaluation of Fr 1C-treated IAV virions. (**A,B**) DMSO and Fr 1C were added to solutions containing purified IAV; the reactions proceeded at 25 °C for 48 h and the viral particles were evaluated using transmission electron microscopy. (**A**) The upper and lower images show the representative DMSO- and Fr 1C-treated viral particles within a single 0.52 µm^2^ field, respectively. Black arrowhead indicates a disrupted viral particle. White arrowheads indicate rough spike proteins. (**B**) The percentage of intact viruses in all viruses within 0.52 µm^2^ fields in which more than 3 viral particles were present was calculated. Thirty fields were assessed in each DMSO- and Fr 1C-treated sample (n = 30 per group). Student’s *t* test was performed to analyze statistical difference; ^**^
*p* < 0.01.

**Figure 4 viruses-12-00699-f004:**
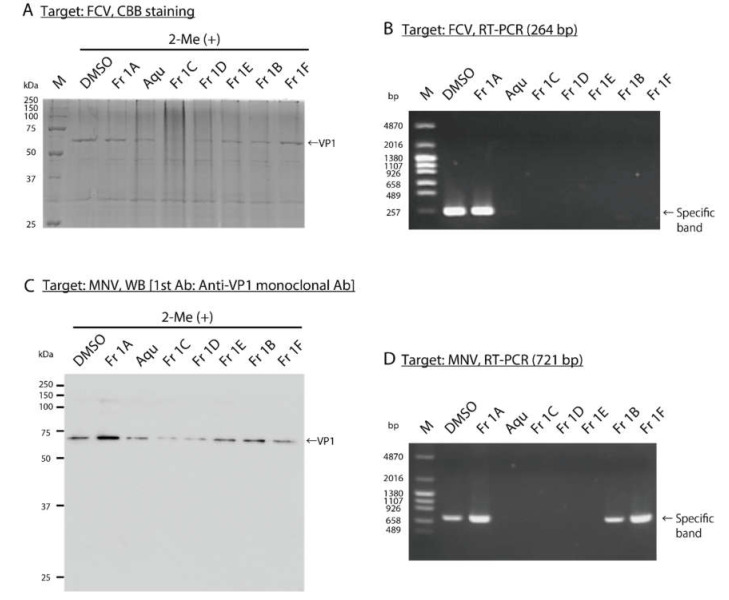
Analysis of the impact of *SS*-derived fractions on FCV and MNV proteins and genomes. (**A, B**) DMSO and *SS*-derived fractions were added to solutions containing purified FCV and the reactions proceeded at 25 °C for (**A**) 18 h followed by SDS–PAGE and CBB staining and (**B**) 12 h followed by RT–PCR using FCV-Primer set which amplifies 264 bp region on FCV gene. M: Marker, Aqu: Aqueous extract. (**C,D**) DMSO and *SS*-derived fractions were added to solutions containing purified MNV and the reactions proceeded at 25 °C for 48 h followed by (**C**) WB to detect MNV VP1 proteins and (**D**) RT–PCR using MNV-F1 and -R1 primers which amplify 721 bp region on MNV gene.

**Figure 5 viruses-12-00699-f005:**
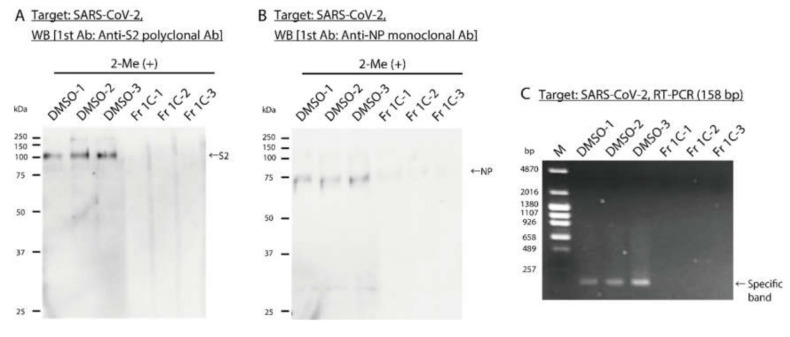
Analysis of the effect of Fr 1C on the SARS-CoV-2 proteins and genome. DMSO and Fr 1C were added to cell culture supernatants containing SARS-CoV-2 and were incubated at 25 °C for 48 h. n = 3 per group. (**A,B**) The images are the results of WB to detect SARS-CoV-2 (**A**) S2 subunit protein and (**B**) NP. (**C**) The image is the result of RT–PCR using NIID_2019-nCoV_N_F2 and R2 primers which amplify 158 bp region on SARS-CoV-2 gene. M: Marker.

**Table 1 viruses-12-00699-t001:** *SS*-derived compounds from each fraction.

Origin (g)	No.	Compound name	Obtained amount (mg)
Fr 1C (5.0 g)	**1**	(2*R*,3*R*)-3’-*O*-*β*-d-(6’’-*O*-galloyl)glucopyranosyloxy-5,7,4’,5’- tetrahydroxyflavanonol	64.5 mg
	**2**	gallocatechin-3-*O*-gallate	88.0 mg
	**3**	dihydromyricetin	2.3 mg
	**4**	epigallocatechin-3-*O*-gallate	4.8 mg
	**5**	catechin	2.3 mg
	**6**	3-*O*-(6”-*O*-galloyl-*β*-d-glucopyranosyl)gallocatechin	21.5 mg
	**7**	6’-*O*-galloyl salidroside	25.7 mg
	**8**	(2*R*,3*R*)-dihydromyricetin 3’-*O*-*β*-d-glucopyranoside	3.2 mg
Fr 1D (5.0 g)	**9**	(2*R*,3*R*)-3’-*O*-*β*-d-(2”,6”-di-*O*-galloyl)glucopyranosyloxy-5,7,4’,5’- tetrahydroxyflavanonol	8.0 mg
	**10**	8-*O*-*β*-d-[6’-*O*-(3’’-*O*-methyl)galloyl]glucopyranosyl-*p*-tyrosol	7.9 mg
	**11**	4-*O*-*β*-d-(6’-*O*-galloyl)glucopyranosyl-(*E*)-*p*-coumaroyl acid	2.9 mg
	**12**	8-*O*-*β*-d-(2’,6’-di-*O*-galloyl)glucopyranosyl-*p*-tyrosol	12.1 mg
	**13**	myricetin	6.2 mg
	**14**	rutin	8.4 mg
	**15**	quercetin 3-*O*-*β*-d-(6”-*O*-galloyl)galactopyranoside	15.6 mg
	**16**	myricetin 3-*O*-*β*-d-galactopyranoside	4.6 mg
	**17**	quercetin 3-*O*-*β*-d-(6”-*O*-galloyl)glucopyranoside	2.5 mg
Fr 1E (3.3 g)	**18**	(2*S*)-3’-*O*-*β*-d-(6”-*O*-galloyl)glucopyranosyloxy-5,7,4’,5’- tetrahydroxyflavanone	7.1 mg
	**19**	(2*R*)-3’-*O*-*β*-d-(6”-*O*-galloyl)glucopyranosyloxy-5,7,4’,5’- tetrahydroxyflavanone	9.1 mg
	**20**	(2*S*)-3’-*O*-*β*-d-(2”,6”-di-*O*-galloyl)glucopyranosyloxy-5,7,4’,5’- tetrahydroxyflavanone	9.8 mg
	**21**	(2*S*)-3’-*O*-*β*-d-[6”-*O*-(3’”-*O*-methyl)galloyl]glucopyranosyloxy-5,7,4’,5’- tetrahydroxyflavanone	4.6 mg
	**22**	naringenin 7-*O*-*β*-d-(6”-*O*-galloyl)glucopyranoside	3.1 mg
	**23**	quercetin	9.2 mg
	**24**	eriodictyol	11.0 mg
	**25**	taxifolin	2.5 mg
	**26**	quercetin 3-*O*-*β*-d-glucopyranoside	1.5 mg
	**27**	quercetin 3-*O*-*α*-l-rhamnopyranoside	2.8 mg
	**28**	gallic acid	10.4 mg
	**29**	4-(4’-hydroxyphenyl)-2-butanone 4’-*O*-*β*-d-(2”,6”-di-*O*- galloyl)glucopyranoside	4.0 mg

**Table 2 viruses-12-00699-t002:** Screening the *SS*-derived fractions for virucidal activity against IAV, FCV, and MNV.

A Target: IAV, concentration of sample: 25 µg/mL.				
Extract and fraction name	[DMSO control ^#)^ - Sample ^#)^] ± SD			
	Reaction time: 24 hrs	6 hrs	10 min	1 min
Aqueous	≥3.33 ± 1.16*	≥3.83 ± 1.16*	≥3.83 ± 0.76*	2.83 ± 0.29**
Fr 1A	0.67 ± 1.16	N.T.	N.T.	N.T.
Fr 1B	≥4.17 ± 0.29**	≥3.83 ± 1.16*	≥4.33 ± 1.26*	0.5 ± 0.5
Fr 1C	≥4.17 ± 0.29**	≥3.83 ± 1.16*	≥4.5 ± 1.32*	≥3.83 ± 0.29**
Fr 1D	≥4.17 ± 0.29**	≥3.83 ± 1.16*	≥3.67 ± 0.76*	1.83 ± 0.29**
Fr 1E	≥4.17 ± 0.29**	≥3.83 ± 1.16*	≥4.17 ± 1.26**	1.1 ± 0.42**
Fr 1F	≥4 ± 0.5**	≥2.1 ± 1.43*	0.5 ± 0.5	N.T.
**B Target: FCV, concentration of sample: 25 µg/mL.**				
**Extract and fraction name**	**[DMSO control ^#)^ - Sample ^#)^] ± SD**			
	**Reaction time:** **48 hrs**	**6 hrs**	**10 min**	**1 min**
Aqueous	≥2.83 ± 0.29**	≥2.83 ± 0.58*	≥2.83 ± 0.29**	≥2.83 ± 0.29**
Fr 1A	≥2.83 ± 0.29**	≥1.75 ± 0.29*	-0.17 ± 0.29	N.T.
Fr 1B	≥2.5 ± 0.5*	≥2.67 ± 0.29**	≥2.67 ± 0.29**	≥3 ± 0.5**
Fr 1C	≥2.83 ± 0.29**	≥2.83 ± 0.58*	≥2.83 ± 0.29**	≥3 ± 0***
Fr 1D	≥2.83 ± 0.29**	≥2.67 ± 0.29**	≥2.83 ± 0.29**	≥3 ± 0***
Fr 1E	≥2.83 ± 0.58*	≥2.67 ± 0.29**	≥2.83 ± 0.29**	≥2.67 ± 0.29**
Fr 1F	≥2.83 ± 0.58*	≥2.67 ± 0.63**	≥2.67 ± 0.29**	≥2.83 ± 0.29**
**C Target: MNV, concentration of sample: 100 µg/mL.**				
**Extract and fraction name**	**[DMSO control ^#)^ - Sample ^#)^] ± SD**			
	**Reaction time:** **48 hrs**	**6 hrs**	**10 min**	**1 min**
Aqueous	1.67 ± 0.12***	1.88 ± 0.25***	0 ± 0.5	N.T.
Fr 1A	0 ± 0.63	N.T.	N.T.	N.T.
Fr 1B	1.83 ± 0.58*	1.88 ± 0.48**	0.33 ± 0.29	N.T.
Fr 1C	≥2.92 ± 0.97***	≥3.13 ± 0.25***	2.17 ± 0.29**	2 ± 0***
Fr 1D	1.67 ± 0.41***	≥2.75 ± 0.65**	2.17 ± 0.29**	1.83 ± 0.29**
Fr 1E	1.4 ± 0.65**	1.38 ± 0.25**	0.67 ± 0.29	N.T.
Fr 1F	0.25 ± 0.5	N.T.	N.T.	N.T.

^#)^: log_10_ TCID_50_/mL. [DMSO control - Sample] indicates the degree of virus titer reduction by each *SS*-derived sample-treatment. N.T.: not tested., * *p* < 0.05, ** *p* < 0.01, *** *p* < 0.001. Light gray or dark gray was used to highlight statistical *vs*. no statistical differences, respectively.

**Table 3 viruses-12-00699-t003:** Screening the *SS*-derived compounds for virucidal activity against IAV, FCV, and MNV.

A Target: IAV, concentration of sample: 25 µg/mL.					
Origin	No.	[DMSO control ^#)^ - Sample ^#)^] ± SD			
		Reaction time: 24 hrs	6 hrs	10 min	1 min
Fr 1C	**1**	1.33 ± 0.76	N.T.	N.T.	N.T.
	**2**	≥4 ± 0.5**	≥2.67 ± 0.76*	≥1.67 ± 0.93**	2.17 ± 0.29**
	**3**	≥4 ± 1*	≥2.5 ± 1.32*	0.92 ± 0.74*	1.83 ± 0.76*
	**4**	≥3.33 ± 1.26*	≥3.83 ± 0.58**	≥2.83 ± 0.29**	1.17 ± 0.29*
	**5**	0.75 ± 0.96	N.T.	N.T.	N.T.
	**6**	0.17 ± 0.76	N.T.	N.T.	N.T.
	**7**	2.25 ± 0.96*	1.5 ± 0.91*	0.5 ± 0.84	N.T.
	**8**	≥2.88 ± 1.32*	2 ± 0.71*	0.33 ± 0.29	N.T.
Fr 1D	**9**	2 ± 0.5*	0.5 ± 0.5	N.T.	N.T.
	**10**	0.83 ± 0.29*	0.67 ± 0.29	N.T.	N.T.
	**11**	1.33 ± 0.29*	0.17 ± 0.29	N.T.	N.T.
	**12**	1.4 ± 0.96*	−0.33 ± 0.29	N.T.	N.T.
	**13**	≥1.75 ± 1.13**	−0.33 ± 0.29	N.T.	N.T.
	**14**	−0.17 ± 1.61	N.T.	N.T.	N.T.
	**15**	−0.17 ± 1.61	N.T.	N.T.	N.T.
	**16**	1.1 ± 0.74*	0.67 ± 0.29	N.T.	N.T.
	**17**	0.33 ± 0.29	N.T.	N.T.	N.T.
Fr 1E	**18**	0.33 ± 0.76	N.T.	N.T.	N.T.
	**19**	0.17 ± 0.56	N.T.	N.T.	N.T.
	**20**	1 ± 1.73	N.T.	N.T.	N.T.
	**21**	0.5 ± 0.5	N.T.	N.T.	N.T.
	**22**	0.67 ± 0.29	N.T.	N.T.	N.T.
	**23**	0.67 ± 0.29	N.T.	N.T.	N.T.
	**24**	0.5 ± 0.32	N.T.	N.T.	N.T.
	**25**	0.5 ± 0.32	N.T.	N.T.	N.T.
	**26**	−0.5 ± 1.32	N.T.	N.T.	N.T.
	**27**	0.17 ± 0.29	N.T.	N.T.	N.T.
	**28**	0.87 ± 1.53	N.T.	N.T.	N.T.
	**29**	0.5 ± 1.32	N.T.	N.T.	N.T.
**B Target: FCV, concentration of sample: 25 µg/mL.**					
**Origin**	**No.**	**[DMSO control ^#)^ - Sample ^#)^] ± SD**			
		**Reaction time:** **48 hrs**	**6 hrs**	**10 min**	**1 min**
Fr 1C	**1**	≥2.5 ± 0.87*	≥2.83 ± 0.29**	0.33 ± 0.29	N.T.
	**2**	≥2.67 ± 0.58*	≥2.67 ± 0.29**	2 ± 0***	≥2.5 ± 0.5*
	**3**	≥2.75 ± 0.5**	≥2.83 ± 0.58*	≥2 ± 0.5*	2 ± 0***
	**4**	≥2.67 ± 0.58*	≥2.83 ± 0.29**	≥2.17 ± 0.29**	≥2.5 ± 0.5*
	**5**	≥2.17 ± 0.58*	≥2 ± 0.5*	−0.33 ± 0.29	N.T.
	**6**	≥2.67 ± 0.58*	1.33 ± 0.29*	0.67 ± 0.29	N.T.
	**7**	≥2.67 ± 0.58*	≥3 ± 0.5**	≥3.5 ± 0.5**	2.33 ± 0.29**
	**8**	≥2.67 ± 0.58*	≥2.67 ± 0.29**	≥3.67 ± 0.29**	1.88 ± 0.29**
Fr 1D	**9**	≥2.5 ± 0.87*	≥2.67 ± 0.29**	1.33 ± 0.29*	1.5 ± 0.5*
	**10**	≥2.25 ± 0.29***	≥1.83 ± 0.58*	−0.17 ± 0.29	N.T.
	**11**	≥2.33 ± 0.58*	≥2.17 ± 0.58*	0 ± 0	N.T.
	**12**	≥2.33 ± 0.58*	0.83 ± 0.58	N.T.	N.T.
	**13**	≥2.67 ± 0.58*	≥2.38 ± 1.03*	−0.17 ± 0.29	N.T.
	**14**	0.83 ± 0.29*	N.T.	N.T.	N.T.
	**15**	≥2.5 ± 0.5*	1 ± 1.32	N.T.	N.T.
	**16**	≥2.33 ± 0.29**	≥1.75 ± 0.87*	−0.17 ± 0.29	N.T.
	**17**	≥2.38 ± 0.63**	0 ± 0	N.T.	N.T.
Fr 1E	**18**	≥2.25 ± 0.65**	0.5 ± 0.71	N.T.	N.T.
	**19**	0.67 ± 1.16	N.T.	N.T.	N.T.
	**20**	≥2.83 ± 0.58*	≥1.8 ± 1.3*	−0.17 ± 0.29	N.T.
	**21**	≥2.25 ± 0.96*	0.17 ± 0.29	N.T.	N.T.
	**22**	≥2.17 ± 0.29**	1.5 ± 0***	−0.67 ± 0.29	N.T.
	**23**	≥1.8 ± 1.3*	≥1.9 ± 0.25*	−0.5 ± 0.5	N.T.
	**24**	≥2.67 ± 0.58*	≥1.7 ± 0.98*	−0.67 ± 0.29	N.T.
	**25**	≥2.67 ± 0.58*	1.13 ± 1.11	N.T.	N.T.
	**26**	≥1.83 ± 0.29**	0.75 ± 0.65	N.T.	N.T.
	**27**	0.83 ± 0.76	N.T.	N.T.	N.T.
	**28**	≥1.5 ± 0.71*	1.67 ± 0.29*	1.83 ± 0.58*	1.33 ± 0.29**
	**29**	≥2.33 ± 0.58*	≥1.9 ± 1.14*	0.67 ± 0.29	N.T.
**C Target: MNV, concentration of sample: 100 µg/mL.**					
**Origin**	**No.**	**[DMSO control ^#)^ - Sample ^#)^] ± SD**			
		**Reaction time:** **48 hrs**	**6 hrs**	**10 min**	
Fr 1C	**1**	−0.16 ± 0.58	N.T.	N.T.	
	**2**	1.4 ± 0.55**	1.25 ± 0.5*	−0.17 ± 0.29	
	**3**	0.5 ± 0	N.T.	N.T.	
	**4**	1.1 ± 0.23***	1.5 ± 0.41**	0 ± 0.5	
	**5**	0 ± 0.5	N.T.	N.T.	
	**6**	0 ± 0	N.T.	N.T.	
	**7**	0.33 ± 0.29	N.T.	N.T.	
	**8**	0.17 ± 0.29	N.T.	N.T.	
Fr 1D	**9**	0.33 ± 0.58	N.T.	N.T.	
	**10**	0.33 ± 0.29	N.T.	N.T.	
	**11**	0 ± 0.5	N.T.	N.T.	
	**12**	−0.63 ± 0.48	N.T.	N.T.	
	**13**	−0.33 ± 0.29	N.T.	N.T.	
	**14**	−0.63 ± 0.48	N.T.	N.T.	
	**15**	−0.63 ± 0.29	N.T.	N.T.	
	**16**	−0.33 ± 0.29	N.T.	N.T.	
	**17**	0.17 ± 0.29	N.T.	N.T.	
Fr 1E	**18**	−0.16 ± 0.58	N.T.	N.T.	
	**19**	0 ± 0.41	N.T.	N.T.	
	**20**	0 ± 0	N.T.	N.T.	
	**21**	0 ± 0.5	N.T.	N.T.	
	**22**	−0.33 ± 0.29	N.T.	N.T.	
	**23**	−0.5 ± 0.5	N.T.	N.T.	
	**24**	−0.33 ± 0.29	N.T.	N.T.	
	**25**	−0.33 ± 0.29	N.T.	N.T.	
	**26**	−0.33 ± 0.29	N.T.	N.T.	
	**27**	0 ± 0	N.T.	N.T.	
	**28**	−0.16 ± 0.29	N.T.	N.T.	
	**29**	−0.67 ± 0.29	N.T.	N.T.	

^#)^: log_10_ TCID_50_/mL; [DMSO control - Sample] indicates the degree of virus titer reduction by each *SS*-derived sample-treatment. N.T.: not tested., * *p* < 0.05, ** *p* < 0.01, *** *p* < 0.001. Light gray or dark gray was used to highlight statistical *vs*. no statistical differences, respectively.

**(A) viruses-12-00699-t004a:** 

Extract and Fraction Name	[DMSO control ^#)^ - Sample ^#)^] ± SD
Reaction time:1 min	30 sec	10 sec
Aqueous	≥2.25 ± 0.29***	N.T.	N.T.
Fr 1A	0 ± 0	N.T.	N.T.
Fr 1B	≥2.17 ± 0.58*	N.T.	N.T.
Fr 1C	≥2.5 ± 0.71**	≥2.17 ± 0.41***	≥2.33 ± 0.41***
Fr 1D	≥2.25 ± 0.65**	N.T.	N.T.
Fr 1E	2 ± 0.41**	N.T.	N.T.
Fr 1F	0.5 ± 0.71	N.T.	N.T.

**(B) viruses-12-00699-t004b:** 

Origin	No	[DMSO Control ^#)^ - Sample ^#)^] ± SD	Origin	No	[DMSO Control ^#)^ - Sample ^#)^] ± SD	Origin	No	[DMSO Control ^#)^ - Sample ^#)^] ± SD
Reaction time:1 min	Reaction time:1 min	Reaction time:1 min
Fr 1C	**1**	0.5 ± 0.5	Fr 1D	**9**	0.63 ± 0.48	Fr 1E	**18**	0.17 ± 0.29
**2**	1.7 ± 0.2***	**10**	0.17 ± 0.29	**19**	0.33 ± 0.58
**3**	0.13 ± 0.63	**11**	0 ± 0.5	**20**	0.33 ± 0.29
**4**	1 ± 0.71*	**12**	0 ± 0.5	**21**	0.33 ± 0.29
**5**	−0.33 ± 0.58	**13**	0 ± 0.87	**22**	0.17 ± 0.29
**6**	0.17 ± 0.29	**14**	−0.17 ± 0.29	**23**	0 ± 0.5
**7**	0.38 ± 0.48	**15**	−0.33 ± 0.58	**24**	−0.33 ± 0.29
**8**	0.5 ± 0.41	**16**	−0.33 ± 0.58	**25**	−0.17 ± 0.29
		**17**	0 ± 0	**26**	−0.5 ± 0.5
				**27**	0 ± 0.5
				**28**	0.25 ± 0.29
				**29**	0.33 ± 0.58

^#)^: log_10_ TCID_50_/mL; [DMSO control - Sample] indicates the degree of virus titer reduction by each *SS*-derived sample-treatment. N.T.: not tested., * *p* < 0.05, ** *p* < 0.01, *** *p* < 0.001. Light gray or dark gray was used to highlight statistical *vs*. no statistical differences, respectively.

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
