# Peer review of "Saxifraga spinulosa-Derived Components Rapidly Inactivate Multiple Viruses Including SARS-CoV-2"

_viruses, 2020, doi:10.3390/v12070699_

Round 1

Reviewer 1 Report

The manuscript by Takeda et al. reports effect of SS derived components to inactivate influenza A virus, feline calicivirus, murine norovirus, and SARS-CoV-2.

As the SARS-CoV-2 is the biosafety level 3 virus, it is unclear where the study for SRS-CoV-2 was performed.

The study evaluated four different viruses, but did not use all the same strategies/approaches for them. Such as EM only for IAV. Is there any reason?

Reviewer 2 Report

The manuscript titled "Saxifraga spinulosa-derived components rapidly inactivate multiple viruses including SARS-CoV-2” investigates the ability Saxifraga spinulosa-derived components to inactivate influenza A virus, SARS-CoV2, feline calicivirus and murine norovirus.

The authors have observed complete inactivation of influenza virus after treatment with 6 of 7 fractions of Saxifraga spinulosa and similar effectiveness against Feline calicivirus. Inactivation of MNV was observed after 6hs with some of the fractions and significant virucidal activity was observed in 5 of 7 fractions against SARS-CoV2. While I appreciate what the authors have done, I am also concern about the need to address the following points:

Cell toxicity controls: For each of these fractions, the cell toxicity at the tested dilution on each cell line should be reported. The lack of virus replication after treatments could be consequence of cell dead given the toxicity of the product on the cells during the incubation period of 2-3 days.

It could also be possible that the Saxifraga spinulosa does not result toxic to the cell, but interfere with virus replication. In that case, it will be important to treat the cell with the Saxifraga spinulosa and after that add the virus to show no interference.

Did the authors neutralize the remained active Saxifraga spinulosa after incubation with the virus and before seeding the cultures?

Was a neutralization control used (where the Saxifraga-spinulosa is pre-mixed with neutralization buffer then virus added for the longest incubation time to ensure that the neutralization was completely effective?

It is unclear which concentrations were applied:

Line 129 The SS-derived fractions at concentrations of 25, 25, 100, and 25 mg/ml were added to solutions

Line 140 The SS-derived samples at 250 mg/ml, 250 mg/ml, and 1 mg/ml were added to solutions of

Line 165 Two SS-derived samples (at 250 mg/ml) or DMSO diluent controls were added to a PBS

Line 170 and 189 The SS-derived samples (250 mg/ml) or DMSO diluent

Tables 2,3 and 4: Please indicate that results summarized hear are virus titer reduction.

At 10 min and 6hs inactivation of IAV with Fr 1D and Fr1E was highly significant according to Table 2. Why the individual components were not evaluated? The same observation applies to MNV and Fr1D and 1E.

Figure 1A shows about 1 log reduction on IAV concentration after treatment with Fr1F. However, Table 2 shows >4 log reduction, almost equal to the other Fr.
